# Modern contraceptive use among sexually active women aged 15–19 years in North-Western Tanzania: results from the Adolescent 360 (A360) baseline survey

Mussa Kelvin Nsanya,[1] Christina J Atchison,[2] Christian Bottomley,[2] Aoife Margaret Doyle,[2] Saidi H Kapiga[1,3]

¹Mwanza Intervention Trials Unit, National Institute for Medical Research, Mwanza, Tanzania
²MRC Tropical Epidemiology Group, London School of Hygiene and Tropical Medicine, London, UK
³Infectious Disease Epidemiology, London School of Hygiene and Tropical Medicine, London, UK

**Correspondence to**
Dr Mussa Kelvin Nsanya;
kelvin.nsanya@gmail.com

## ABSTRACT

**Objectives** To describe differences in modern contraceptive use among adolescent women aged 15–19 years according to their marital status and to determine factors associated with modern contraceptive use among sexually active women in this population.

**Design** Cross-sectional analysis of Adolescent 360 evaluation baseline survey.

**Setting** The 15 urban and semiurban wards of Ilemela district, Mwanza region, North-Western Tanzania.

**Participants** Adolescent women aged 15–19 years who were living in the study site from August 2017 to February 2018 and who provided informed consent. Women were classified as married if they had a husband or were living as married. Unmarried women were classified as sexually active if they reported having sexual intercourse in the last 12 months.

**Outcome measure** Prevalence of modern contraceptive among adolescent women aged 15–19 years.

**Results** Data were available for 3511 women aged 15–19 years, of which 201 (5.7%) were married and 744 (22.5%) were unmarried-sexually active. We found strong evidence of differences in use of modern contraceptive methods according to marital status of adolescent women. Determinants of modern contraception use among unmarried-sexually active women were increasing age, increasing level of education, being in education, hearing of modern contraception from interpersonal sources or in the media in the last 12 months, perceiving partner and/or friends support for contraceptive use, as well as higher knowledge and self efficacy for contraception.

**Conclusions** Sexual and reproductive health programmes aiming to increase uptake of modern contraceptives in this population of adolescent women should consider the importance of girl's education and social support for contraceptive use particularly among unmarried-sexually active women.

## INTRODUCTION

Globally, ~16 million women aged 15–19 years give birth each year and 95% of these births take place in low/middle-income countries.[1 2] The global community, through the Family Planning 2020 (FP2020) initiative, is committed to increase new contraceptive

### Strengths and limitations of this study

► We focused on adolescent women aged 15–19 years, a population that is often excluded or underrepresented in most studies on modern contraception
► The probability sampling approach we used allows generalisation of the findings of this study to the wider population of adolescent women aged 15 - 19 years living in the urban and semiurban wards of Ilemela district.
► The main limitation is the cross-sectional design which make temporal causal relationships hard to establish.
► Due to small sample size, we were unable to identify determinants of contraceptive use among married adolescent women.
► We did not collect data on intentions to get pregnant hence we have no data on this potential explanatory variable.

users to 120 million in 69 developing countries (including Tanzania) by 2020.[3] This initiative would also support the objectives of the United Nation's Sustainable Development Goal (SDG) 3 on health and well-being for all and SDG 5 on gender equality which also embodies sexual and reproductive health at the heart of global efforts to sustainable development particularly in low/middle-income countries.[4]

Modern contraceptive use remains unacceptably low in sub-Saharan Africa despite increasing awareness and knowledge about contraception.[5 6] For instance, 69% and 59% of the women aged 15–19 years in UK and USA, respectively, report using a modern contraceptive at the last time they had sexual intercourse compared with 12% in Mali and 21% in Tanzania.[7] The low uptake of modern contraceptives particularly among women aged 15–19 years contributes significantly to high rates of adolescent pregnancies and poor health outcomes including maternal

morbidity and mortality, and neonatal and under-five child mortality.[1 5 8] In addition, there are other severe social and economic consequences to adolescent women, their families and whole society including not reaching their potential for educational achievement, and not getting a paid job which usually leads into a vicious cycle of poverty.[1 9]

Most studies to date have focused on the factors that prevent women of reproductive age (15–49 years) from using modern contraceptives.[5] In such studies, adolescent women are usually underrepresented despite facing disproportionate medical, social and economic impact of unintended pregnancies.[1] In order for the goals of FP2020 and SDG 3 and 5 to be achieved, more information is required from studies which examine factors associated with contraceptive use among adolescent women particulary in developing countries, including Tanzania. Moreover, demand for and access to modern contraceptives among adolescent women aged 15–19 years are known to differ with the women's marital status.[10] It is therefore important to describe differences according to marital status to optimise access and use of modern contraception.

Adolescents 360 (A360) is an initiative being rolled out across Ethiopia, Nigeria and Tanzania, aiming to increase uptake of voluntary modern contraception among sexually active women aged 15–19 years.[11] Using baseline survey data collected as part of A360 programme evaluation, we describe differences in modern contraceptive use among adolescent women aged 15–19 years according to their marital status; and determine factors associated with modern contraceptive use among unmarried-sexually active women in this population in Mwanza, Tanzania.

## METHODS
### Study design and settings
Between August 2017 and February 2018, we conducted a cross-sectional baseline survey among women aged 15–19 years in Mwanza city, Tanzania. The survey was part of a comprehensive outcome evaluation to assess the impact of the A360 programme on a number of sexual and reproductive health outcomes, primarily uptake of voluntary modern contraception among sexually active women aged 15–19 years.[11]

In Tanzania, A360 is being implemented in 16 administrative regions. This survey was conducted in 15 urban and semiurban wards of Ilemela district, Mwanza region.[11] Ilemela district covers the northern part of Mwanza city and is comprised of 19 wards, of which 4 are rural wards, 6 are semiurban and 9 are urban wards. Each ward is administratively divided into a number of neighbourhoods, called 'streets'.

This survey (which is part of the intervention evaluation) was set in urban and semiurban wards of Ilemela district because Population Services International (PSI), the A360 intervention implementers in Tanzania, are focusing their efforts in more densely populated areas.

### Study population
Women were included in the study if they were 15–19 years old; living in the study sites at the time of the survey; and voluntarily provided informed consent. Women were classified as married if they reported that they had a husband or were living as married with a cohabiting male partner.

### Informed consent
Written informed consent was obtained from all participants. A parental consent waiver was granted for unmarried women aged 15–17 years, given the sensitive nature of the topics discussed. Married women under 18 years of age were considered emancipated and did not require parental consent in addition to their own voluntary consent.

### Sampling strategy and sample size
A cluster sampling design was used. The primary sampling unit for the survey was a street, the smallest administrative unit similar to a neighbourhood or a localised and delineated group of people. All 15 urban or semiurban wards of Ilemela district were included in the survey. Each ward has an estimated 8–10 streets. A simple random sample of 34 streets was selected across the 15 urban and semiurban wards of Ilemela district. As per study protocol, in the first eight streets, we randomly selected 50 Global Positioning System (GPS) coordinates using ArcGIS software V.9.3. All households whose front door was located within a radius of 20 m around the GPS coordinates were visited and all eligible consenting women aged 15–19 years residing in these households were invited to be interviewed. This approach was considered due to lack of detailed lists of households in the streets which could serve as a sampling frame.[12] As fewer eligible women than predicted were surveyed in each cluster using this sampling strategy, in the remaining 26 streets we visited all households and administered the questionnaire to all eligible and consenting women aged 15–19 years. Our target sample size for the baseline survey was 3314 women aged 15–19 years. Sample size and power calculations were conducted ahead of deciding on the number of streets to sample. The estimated sample size had 90% power to detect a 6% increase in prevalence of modern contraceptive (mCRP) use in presence of A360 intervention for 24 months.[11]

If potentially eligible participants were not available at the first visit, two further revisits were made to attempt to hold interviews.

### Participants and public involvement
We sought permission from local government authorities in the wards where the study took place as well as from individual participants prior to enrolment. Additionally, we have communicated the A360 baseline survey report to local government officials in Ilemela district, Mwanza

## Tool for baseline survey

The questionnaire was adapted from various research instruments that have been used in the target countries including the Tanzania Demographic and Health Survey (DHS)[13] and FP2020 survey instruments.[3] Questionnaires were administered face-to-face by female interviewers aged between 18 and 26 years using preprogrammed tablet computers.[11]

The questionnaire had three components: (1) sociodemographic characteristics (2) fertility characteristics and preferences, and (3) contraceptive knowledge, attitudes and practices. Only respondents who reported sexual intercourse in the last 12 months were considered sexually active hence asked questions about contraceptive use.[11]

## Study outcome

The mCPR among married women aged 15–19 years was defined as per the DHS definition[14]:

$$\frac{\text{Number of married } 15-19-\text{year}-\text{old women reporting use of modern contraceptives at the time of the survey}}{\text{Number of married } 15-19-\text{year}-\text{old women}}$$

mCPR among unmarried-sexually active women aged 15–19 years was defined as follows:

$$\frac{\text{Number of unmarried}-\text{sexually active* women aged } 15-19-\text{year}-\text{old reporting use of modern contraceptives at the time of the survey}}{\text{Number of unmarried}-\text{sexually active* unmarried women aged } 15-19-\text{year}-\text{old}}$$

*self-reported that they were sexually active in the 12 months prior to the survey.

Modern contraception was defined to include the following: male and female sterilisation, contraceptive implants, intrauterine contraceptive devices, injectables, contraceptive pill/oral contraceptives, emergency contraceptive pill, male condom, female condom, standard days method (SDM), lactational aenorrhoea method, diaphragm, spermicides, foams and jelly.[15]

## Statistical analysis

Data analysis was conducted in Stata V.15. We used sampling weights and robust SEs to account for the clustered sampling design.

Descriptive data analysis was done for both married and unmarried women. Logistic regression was performed for unmarried-sexually active women only due to small sample size for married women. We obtained ORs for the association of each explanatory variable with use of modern contraception. Wald tests adjusted for the clustered sampling design were used at each step of the analysis. The associations between mCPR and age and between mCPR and religion were not adjusted for other explanatory variables. Age and religion were considered a priori potential confounders for the associations between mCPR and highest education level achieved, currently being in education, type of area of residence (urban or semiurban) and socioeconomic position. The remaining explanatory variables with p value <0.2 in the univariate analysis, were investigated one-by-one in multivariate regression models adjusted for age, religion, highest education level achieved, currently being in education and socioeconomic position. Variables with p value <0.05 in the adjusted analysis were considered to be associated with mCPR. This strategy allowed us to assess the effect of variables adjusted for distal a priori potential confounders.

### Socioeconomic position

Socioeconomic position was created from a series of questions about household items, dwelling materials and access to a bank account. The variable was generated using the 'Tanzania Equity Tool' which uses different weights attached to each answer to create a composite score which was then split into quintiles according to the national thresholds.[16]

### Knowledge about contraception

Knowledge about contraception was assessed through the respondents affirmative report to the following five questions: (1) preventing unintended pregnancies is a benefit of contraception, (2) preventing abortions is a benefit of contraception, (3) some contraceptive methods reduce sexually transmitted infections/HIV, (4) modern contraception can help with child spacing and (5) using modern contraception can allow a woman to complete her education, take up better economic opportunities and fulfil her potential.

### Holding misconceptions

Holding misconceptions were assessed by asking respondents whether they agreed with the following four statements: (1) use of a long-acting reversible contraceptive can make adolescent women permanently infertile, (2) changes to normal menstrual bleeding patterns, which is caused by some contraceptives, are harmful to health, (3) modern contraceptives can make adolescent women permanently fat and (4) adolescent women who use family planning/birth spacing may become promiscuous.

### Self-efficacy for contraception

Self-efficacy for contraception was assessed through four questions relating to the woman's ability to access and use contraception: (1) felt able to start a conversation with her partner about contraception, (2) felt able to use a method of contraception even if her partner did not want her to, (3) felt able to obtain information on contraception services and products if she needed to and (4) felt able to obtain a contraceptive method if she decided to use one.

Variables for contraception knowledge, holding misconceptions and self-efficacy were created as scores from 0 to 5 for knowledge, and 0 to 4 for holding misconceptions and self-efficacy based on the overall score for each individual statement in each category. A score of 1 was given if the respondent agreed with the statement and 0 if she disagreed or answered '*don't know*'. A maximum score of 5 for knowledge and 4 for self-efficacy would indicate that the respondent correctly agreed with all five knowledge statements and felt able to achieve all four self-efficacy behaviours. A maximum score of 4 for holding misconceptions would be interpreted as believing all four myth statements about contraception.

## RESULTS

A total of 14138 households were identified and 99.6% were interviewed to obtain information on household members. A total of 5121 potentially eligible women aged 15–19 years were identified from 3963 households (28.1% of all interviewed households); 68.6% (3511) of potentially eligible women were interviewed, of whom 5.7% (201/3511) were married. Overall, 22.5% (744/3310) of unmarried women had been sexually active during the 12 months preceding the survey. The most common reason for a potentially eligible woman not interviewed was being absent or unavailable at home after a maximum of three visits mainly due to attending school.

In table 1, we present the characteristic of married and unmarried-sexually active women aged 15–19 years in the study population. The median age was 19 years for married women and 18 years for unmarried-sexually active women. The majority of respondents were Christian (married: 79.7%, unmarried: 84.5%) and were not currently pursuing educational training (married: 99.0%, unmarried: 79.6%). The highest level of education achieved by majority of married women was primary education (65.2%), while the majority of unmarried-sexually active women had achieved secondary level education (52.7%). Most respondents had moderate knowledge about contraception (married: 56.2%, unmarried: 62.4%) and a moderate self-efficacy for contraception (married: 53.0%, unmarried: 56.7%).

In table 2, we present the prevalence of contraceptive use among women aged 15–19 years in Mwanza by their marital status. Overall, 19.4% of married respondents and 48.7% of unmarried-sexually active respondents were using a modern method of contraception. Of those reporting using a modern method of contraception, implants (38.5%) were the most widely used method by married women, followed by injectables (23.1%) and SDM (15.4%). Male condoms (71.6%) were the most widely used modern contraceptive method by unmarried-sexually active women, followed by SDM (15.8%). Overall, there was strong evidence of differences in use of modern contraceptive methods according to marital status of adolescent women (table 2).

In table 3, we present factors associated with use of modern contraceptive methods among unmarried-sexually active women aged 15–19 years in Mwanza, Tanzania.

### Sociodemographic factors

Age, religion, level of education, being in an educational programme and socioeconomic position were all associated with use of modern contraceptive methods in univariate analysis (p value <0.2). The odds of using modern contraception increased with age (adjusted OR (adjOR) 1.2, 95% CI 1.1 to 1.4). Following adjustment for age and religion, there was strong evidence that unmarried-sexually active women who had reached university level education were three times more likely to use modern contraception compared with those with no education (adjOR 3.0, 95% CI 1.0, 9.0, p value 0.004) and those who were not in education had significantly lower odds of using modern contraception compared with those in education (adjOR 0.52, 95% CI 0.36 to 0.75).

### Exposure to information about contraception

Hearing about contraception in the media in the last 12 months or from an interpersonal source, and knowing of a place or person from whom respondent would feel comfortable accessing contraception were significantly associated with using modern contraception in univariate analysis (p value <0.2).

After adjusting for sociodemographic variables, the odds of using modern contraception were significantly lower for unmarried-sexually active women who had not heard about contraception in the media in the last 12 months (adjOR 0.58, 95% CI 0.35 to 0.95) or from interpersonal sources (adjOR 0.61, 95% CI 0.42 to 0.90) compared with those that had heard this information.

### Social network factors

Perceived support from partner, mother and friends for using contraception were associated with using modern contraception in univariate analysis (p value <0.2). After adjusting for sociodemographic variables, the odds of using modern contraception were lower for unmarried-sexually active women who perceived that their partners did not support their use of contraception (adjOR 0.29, 95% CI 0.21 to 0.42) and similarly for those who perceived lack of support from their friends (adjOR 0.55, 95% CI 0.34 to 0.88) compared with those who perceived their support.

### Individual knowledge, attitudes and behaviour factors

Knowledge about contraception, holding misconceptions against contraception, self-efficacy for contraception and number of living children were associated with using modern contraception in univariate analysis (p value <0.2). After adjusting for demographic variables, the odds of using modern contraception were significantly lower in unmarried-sexually active

**Table 1** Characteristics of married and unmarried-sexually active women aged 15–19 years in Mwanza, Tanzania*

| Characteristic | Married, N=201 N (%) | Unmarried, N=744 N (%) | P value |
|---|---|---|---|
| **Sociodemographic factors** | | | |
| Age (years) | | | |
| 15 | 2 (1.0) | 62 (8.3) | <0.0001 |
| 16 | 4 (2.0) | 82 (11.0) | |
| 17 | 24 (11.9) | 161 (21.6) | |
| 18 | 69 (34.3) | 199 (26.8) | |
| 19 | 102 (50.8) | 240 (32.3) | |
| Age (years)† | 19 (18,19) | 18 (17–19) | <0.0001 |
| **Religion** | | | |
| Catholic | 61 (30.4) | 309 (41.5) | 0.04 |
| Protestant/other Christian | 99 (49.3) | 320 (43.0) | |
| Muslim | 38 (18.9) | 112 (15.1) | |
| No religion | 3 (1.5) | 3 (0.40) | |
| Highest level of education achieved | | | |
| No education | 15 (7.5) | 21 (2.8) | <0.0001 |
| Primary education | 131 (65.2) | 320 (43.0) | |
| Secondary education | 55 (27.4) | 392 (52.7) | |
| University education | 0 | 11 (1.5) | |
| **Currently in educational training programme** | | | |
| Yes | 2 (1.0) | 152 (20.4) | <0.0001 |
| No | 199 (99.0) | 592 (79.6) | |
| Type of area of residence | | | |
| Semiurban | 85 (42.3) | 290 (39.0) | 0.43 |
| Urban | 116 (57.7) | 454 (61.0) | |
| **Socioeconomic level** | | | |
| Lowest quintile | 38 (22.1) | 87 (15.5) | 0.0002 |
| Second lowest quintile | 55 (32.0) | 131 (23.4) | |
| Middle quintile | 31 (18.0) | 82 (14.6) | |
| Second highest quintile | 36 (20.9) | 125 (22.3) | |
| Highest quintile | 12 (7.0) | 135 (24.1) | |
| **Exposure to information about contraception** | | | |
| Heard about contraception in the media in last 12 months | | | |
| Yes | 59 (29.4) | 309 (41.5) | <0.0001 |
| No | 142 (70.7) | 435 (58.5) | |
| Heard about contraception from health sector sources in last 12 months | | | |
| Yes | 122 (60.7) | 213 (28.6) | <0.0001 |
| No | 79 (39.3) | 531 (71.4) | |
| **Heard about contraception from interpersonal sources in last 12 months** | | | |
| Yes | 100 (49.8) | 487 (65.5) | 0.0001 |
| No | 101 (50.3) | 257 (34.5) | |
| Knows a place where or person from whom she would feel comfortable accessing contraception | | | |
| Yes | 113 (61.1) | 400 (53.8) | 0.14 |
| No | 72 (38.9) | 343 (46.2) | |
| **Social networks** | | | |
| Perceives that partner supports her using contraception | | | |

| Characteristic | Married, N=201 | Unmarried, N=744 | P value |
|---|---|---|---|
| | N (%) | N (%) | |
| Yes | 116 (62.7) | 430 (60.2) | 0.04 |
| No | 45 (24.3) | 140 (19.6) | |
| Do not know | 24 (13.0) | 144 (20.2) | |
| **Perceives that mother supports her using contraception** | | | |
| Yes | 89 (50.9) | 299 (42.4) | 0.03 |
| No | 53 (30.3) | 190 (26.9) | |
| Do not know | 33 (18.9) | 217 (30.7) | |
| **Perceives that friends supports her using contraception** | | | |
| Yes | 85 (46.2) | 430 (58.3) | 0.02 |
| No | 38 (20.7) | 100 (13.6) | |
| Do not know | 61 (33.2) | 207 (28.1) | |
| **Individual knowledge, attitudes and behaviours** | | | |
| Knowledge about contraception score‡ 0–1 | 21 (10.5) | 37 (5.0) | 0.02 |
| 2–3 | 67 (33.3) | 243 (32.7) | |
| 4–5 | 113 (56.2) | 464 (62.4) | |
| **Misconceptions about contraception score§** | | | |
| 0–1 | 83 (41.3) | 258 (34.7) | |
| 2–3 | 75 (37.3) | 375 (50.4) | |
| 4 | 43 (21.4) | 111 (14.9) | 0.04 |
| **Self-efficacy for contraception score ¶** | | | |
| 0–1 | 15 (8.1) | 57 (7.7) | |
| 2–3 | 72 (38.9) | 265 (35.7) | |
| 4 | 98 (53.0) | 421 (56.7) | 0.67 |
| **Timing of most recent sexual activity** | | | |
| Within last week | 86 (42.8) | 48 (6.5) | |
| Within last month | 52 (25.9) | 207 (27.8) | |
| Within last year | 63 (31.3) | 489 (65.7) | <0.0001 |
| **No of living children** | | | |
| No children | 97 (48.3) | 638 (85.8) | |
| One child or more | 104 (51.7) | 106 (14.3) | <0.0001 |

\*The figures refers to N (%). Numbers and percentages may not match exactly because the analysis used sampling weights to account for the sampling design.
†Median (IQR).
‡Scored based on the responses to the following five questions: (1) preventing unintended pregnancies is a benefit of contraception, (2) preventing abortions is a benefit of contraception, (3) some contraceptive methods reduce sexually transmitted infections/HIV, (4) modern contraception can help with child spacing and (5) using modern contraception can allow a woman to complete her education, take up better economic opportunities and fulfil her potential.
§Scored based on the responses to the following four questions: (1) use of a long-acting reversible contraceptive can make adolescent women permanently infertile, (2) changes to normal menstrual bleeding patterns, which is caused by some contraceptives, are harmful to health, (3) modern contraceptives can make adolescent women permanently fat, and adolescent women who use family planning/birth spacing may become promiscuous.
¶Scored based on the responses to the following four questions: (1) felt able to start a conversation with her partner about contraception, (2) felt able to use a method of contraception even if her partner did not want her to, (3) felt able to obtain information on contraception services and products if she needed to and (4) felt able to obtain a contraceptive method if she decided to use one.

women with one or more living children compared with those without a child (adjOR 0.57, 95% CI 0.39 to 0.85). The odds of using modern contraception increased with higher knowledge about contraception (adjOR 2.4, 95% CI 1.2 to 4.6) and higher self-efficacy for using contraception (adjOR 2.4, 95% CI 1.5 to 4.1).

## DISCUSSION

In this paper, we describe differences in modern contraceptive use among adolescecnt women aged 15–19 years taking part in the A360 evaluation baseline survey according to their marital status. We also present determinants for modern contraception among unmarried-sexually active women aged 15–19 years who were enrolled in this study.

**Table 2**  Prevalence of contraceptive use among women aged 15–19 years in Mwanza, Tanzania by marital status*

| Characteristic | Married, N=201† | Unmarried, N=744† | P value |
|---|---|---|---|
| Any method | 20.4 (13.9–28.9) | 50.7 (47.7–53.6) | <0.0001 |
| Any modern method‡ | 19.4 (13.4–27.3) | 48.7 (45.8–51.5) | <0.0001 |
| Any traditional method | 1.0 (0.22–4.4) | 2.0 (1.3–3.0) | |
| Not currently using | 79.6 (71.2–86.1) | 49.3 (46.4–52.3) | |
| Total | 100.0 | 100.0 | |
| Modern method | | | |
| Implant | 38.5 (21.4–58.9) | 4.4 (2.4–7.9) | <0.0001 |
| IUCD | 7.7 (2.5–21.5) | 0.28 (0.03–2.3) | |
| Injectables | 23.1 (10.8–42.7) | 5.8 (3.8–8.7) | |
| Contraceptive pill/oral contraceptives | 2.6 (0.31–18.0) | 0.55 (0.13–2.4) | |
| Emergency pill | 0 | 0.55 (0.12–2.4) | |
| Male condom | 7.7 (2.4–22.2) | 71.6 (66.9–75.8) | |
| Standard days method | 15.4 (5.2–37.6) | 15.8 (12.1–20.3) | |
| Other modern method | 5.1 (1.4–16.6) | 1.1 (0.37–3.2) | |
| Total | 100.0 | 100.0 | |

*Figures are % (95% CI) taking into account sampling weights to account for the sampling design.
†Unmarried girls who report sexually activity in last 12 months; all married girls in Ilemela district.
‡Modern methods include female sterilisation, male sterilisation, contraceptive pill (oral contraceptives), IUCD, injectables (Depo-Provera), implants (Norplant), female condom, male condom, diaphragm, contraceptive foam and contraceptive jelly, LAM, SDM, cycle beads.
IUCD, intrauterine contraceptive devices; LAM, lactational amenorrhoea method; SDM, standard days method.

We found more married women (99.0%) than their unmarried counterparts (79.6%) were out of the formal educational system. In contrast, more unmarried women (52.7%) than married women (27.4%) had achieved secondary school education. These findings highlights an inverse relationship between child marriage, which is prevalent in Tanzania,[17 18] and education achievements. Child marriage is known to impact negatively on educational attainment by adolescent women and perpetuates a vicious cycle of poverty at individual, family and community levels.[1 9] However, child marriage is increasingly acknowledged as a violation of girls' human rights which must be protected by family, community and government authorities.[18]

In this study, although both married and unmarried-sexually active women showed similar levels of knowledge about contraception (56.2% vs 62.4%) and self-efficacy for modern contraception use (53.0% vs 56.7%), the proportion of modern contraceptive users was far lower among married women when compared with unmarried women (19.4% vs 48.7%). Generally, in sub-Saharan Africa, modern contraception use has remained low despite the rising awareness and knowledge.[5] However, the observed disparities could be pointing to presence of other factors that may have a substantial inhibitive role to contraceptive uptake particularly among married adolescent women. One of those determinants is the social pressure for adolescent women to prove their fertility immediately after marriage,[10 19] which has also been shown to be used as an anchor to win over husband's respect and stabilise marital relationship.[5 20] In this context, proving own fertility, carries more weight and higher priority over higher contraceptive knowledge and self efficacy in deciding whether or not to use modern contraception.

The most commonly used contraceptive methods were implants (38.5%) among married women and male condoms (71.6%) among unmarried-sexually active women. In a context of held misconceptions particularly against hormonal-based contraceptives for their perceived interference on fertility, this finding might reflect an attempt by unmarried-sexually active women to preserve their fertility by opting for non-hormonal based and/or non-invasive methods. Other studies have reported that some adolescent women have chosen unsafe clandestine abortions over hormonal-based contraception.[5 21] This finding may also reflect that unmarried-sexually active adolescent women could have better access to male condoms compared with other methods. In addition, it could reflect the level of exposure and frequency of sexual intercourse, and the type of sexual relationships for unmarried adolescent women which may not require a long-term contraception method. Despite the limitation of condoms being male controlled, the double role of condoms in preventing unintended pregnancies and sexually transmitted infections including HIV, stresses the need for making condoms more accessible and advocating for their proper and regular use among unmarried-sexually active adolescent women.[7]

Modern contraception use among unmarried-sexually active women in the study population was associated with

**Table 3** Factors associated with modern contraception use among unmarried-sexually active women aged 15–19 years in Mwanza, Tanzania, (N=744)*

| Exposure category | No | Prevalence, n (%) | Unadjusted OR (95% CI) | P value | Adjusted OR (95% CI) | P value |
|---|---|---|---|---|---|---|
| **Sociodemographic factors** | | | | | | |
| Age (years) | | | | | | |
| 15 | 62 | 20 (32.3) | | | | |
| 16 | 82 | 30 (36.6) | | | | |
| 17 | 161 | 79 (49.1) | | | | |
| 18 | 199 | 107 (53.8) | | | | |
| 19 | 240 | 126 (52.5) | | | | |
| Per year increase | | | 1.2 (1.1 to 1.4) | 0.01 | | |
| Religion | | | | | | |
| Catholic | 309 | 145 (46.9) | 1 | 0.1 | | |
| Protestant/other Christian | 320 | 149 (46.6) | 0.99 (0.77 to 1.3) | | | |
| Muslim | 112 | 66 (58.9) | 1.6 (1.1 to 2.4) | | | |
| No religion | 3 | 2 (66.7) | 2.3 (0.50 to 10.2) | 0.1 | | |
| Highest educational achieved† | | | | | | |
| No education | 21 | 7 (33.3) | 1 | | 1 | |
| Primary | 320 | 127 (39.7) | 1.3 (0.39 to 4.5) | | 1.4 (0.40 to 4.7) | |
| Secondary | 392 | 221 (56.4) | 2.6 (0.83 to 8.0) | | 2.5 (0.78 to 8.1) | |
| University | 11 | 7 (63.6) | 3.5 (1.2 to 9.8) | 0.0006 | 3.0 (1.0 to 9.0) | 0.004 |
| Currently in educational training† | | | | | | |
| Yes | 152 | 89 (58.6) | 1 | 0.008 | 1 | 0.002 |
| No | 592 | 273 (46.1) | 0.61 (0.43 to 0.86) | | 0.52 (0.36 to 0.75) | |
| Type of area of residence† | | | | | | |
| Semiurban | 290 | 140 (48.3) | 1 | 0.82 | | |
| Urban | 454 | 222 (48.9) | 1.0 (0.82 to 1.3) | | | |
| Socioeconomic level† | | | | | | |
| Lowest quintile | 87 | 36 (41.4) | 1 | | 1 | |
| Second lowest quintile | 131 | 62 (47.3) | 1.3 (0.71 to 2.3) | | 1.2 (0.70 to 2.1) | |
| Middle quintile | 82 | 43 (52.4) | 1.6 (0.80 to 3.0) | | 1.5 (0.72 to 3.1) | |
| Second highest quintile | 125 | 59 (47.2) | 1.3 (0.70 to 2.3) | | 1.2 (0.60 to 2.3) | |
| Highest quintile | 135 | 79 (58.5) | 2.0 (1.1 to 3.6) | 0.08 | 1.9 (1.1 to 3.4) | 0.09 |
| **Exposure to information about contraception** | | | | | | |
| Heard about contraception in the media in last 12 months‡ | | | | | | |
| Yes | 309 | 174 (56.3) | 1 | | 1 | |
| No | 435 | 188 (43.2) | 0.59 (0.42 to 0.83) | 0.004 | 0.58 (0.35 to 0.95) | 0.03 |
| Heard about contraception from health sector sources in last 12 months | | | | | | |
| Yes | 213 | 101 (47.4) | 1 | | | |
| No | 531 | 261 (49.2) | 1.1 (0.71 to 1.6) | 0.73 | | |
| Heard about contraception from interpersonal sources in last 12 months‡ | | | | | | |
| Yes | 487 | 261 (53.6) | 1 | | 1 | |
| No | 257 | 101 (39.3) | 0.56 (0.40 to 0.78) | 0.002 | 0.61 (0.42 to 0.90) | 0.01 |
| Know of a place where or person from whom she would feel comfortable accessing contraception‡ | | | | | | |
| Yes | 400 | 213 (53.3) | 1 | | 1 | |
| No | 343 | 149 (43.4) | 0.67 (0.50 to 0.92) | 0.01 | 0.69 (0.46 to 1.0) | 0.07 |
| **Social Networks** | | | | | | |
| Perceives that partner supports her using contraception‡ | | | | | | |
| Yes | 430 | 264 (61.4) | 1 | | 1 | |
| No | 140 | 40 (28.6) | 0.25 (0.18 to 0.35) | | 0.29 (0.21 to 0.42) | |
| Do not know | 144 | 53 (36.8) | 0.37 (0.24 to 0.55) | <0.0001 | 0.32 (0.20 to 0.52) | <0.0001 |

Continued

**Table 3** Continued

| Exposure category | No | Prevalence, n (%) | Unadjusted OR (95% CI) | P value | Adjusted OR (95% CI) | P value |
|---|---|---|---|---|---|---|
| Perceives that mother supports her using contraception‡ | | | | | | |
| Yes | 299 | 160 (53.5) | 1 | | 1 | |
| No | 190 | 89 (46.8) | 0.77 (0.53 to 1.1) | | 0.87 (0.56 to 1.4) | |
| Do not know | 217 | 97 (44.7) | 0.70 (0.54 to 0.92) | 0.05 | 0.73 (0.48 to 1.1) | 0.32 |
| Perceives that friends supports her using contraception‡ | | | | | | |
| Yes | 430 | 240 (55.8) | 1 | | 1 | |
| No | 100 | 44 (44.0) | 0.62 (0.45 to 0.86) | | 0.55 (0.34 to 0.88) | |
| Do not know | 207 | 76 (36.7) | 0.46 (0.33 to 0.63) | 0.0004 | 0.42 (0.29 to 0.61) | 0.0004 |
| Individual knowledge, attitudes and behaviours | | | | | | |
| Knowledge about contraception‡§ | | | | | | |
| 0–1 | 37 | 12 (32.4) | 1 | | 1 | |
| 2–3 | 243 | 104 (42.8) | 1.6 (0.98 to 2.5) | | 1.9 (1.0 to 3.4) | |
| 4–5 | 464 | 246 (53.0) | 2.4 (1.4 to 4.0) | 0.01 | 2.4 (1.2 to 4.6) | 0.05 |
| Misconceptions about contraception‡¶ | | | | | | |
| 0–1 | 258 | 114 (44.2) | 1 | | 1 | |
| 2–3 | 375 | 185 (49.3) | 1.2 (0.80 to 1.9) | | 0.93 (0.58 to 1.5) | |
| 4 | 111 | 63 (56.8) | 1.7 (0.96 to 2.9) | 0.19 | 1.4 (0.82 to 2.4) | 0.34 |
| Self-efficacy for contraception‡** | | | | | | |
| 0–2 | 117 | 28 (23.9) | 1 | | 1 | |
| 3–4 | 626 | 334 (53.4) | 3.6 (2.4 to 5.5) | <0.0001 | 2.4 (1.5 to 4.1) | 0.002 |
| Timing of most recent sexual activity | | | | | | |
| Within last week | 48 | 24 (50.0) | 1 | | | |
| Within last month | 207 | 110 (53.1) | 1.1 (0.58 to 2.2) | | | |
| Within last year | 489 | 228 (46.6) | 0.87 (0.47 to 1.6) | 0.42 | | |
| No of living children‡ | | | | | | |
| No children | 638 | 321 (50.3) | 1 | | 1 | |
| One child or more | 106 | 41 (38.7) | 0.62 (0.44 to 0.89) | 0.01 | 0.57 (0.39 to 0.85) | 0.008 |

P value from design based Wald test.

*Numbers and percentages may not match exactly because the analysis used sampling weights to account for the sampling design.

†Adjusted ORs: adjusted for age and religion.

‡Adjusted ORs: adjusted for age, religion, highest education level achieved, currently in education and socioeconomic position.

§Scored based on the responses to the following five questions: (1) preventing unintended pregnancies is a benefit of contraception, (2) preventing abortions is a benefit of contraception, (3) some contraceptive methods reduce sexually transmitted infections/HIV, (4) modern contraception can help with child spacing and (5) using modern contraception can allow a woman to complete her education, take up better economic opportunities and fulfil her potential.

¶Scored based on the responses to the following four questions: (1) use of a long-acting reversible contraceptive can make adolescent women permanently infertile, (2) changes to normal menstrual bleeding patterns, which is caused by some contraceptives, are harmful to health, (3) modern contraceptives can make adolescent women permanently fat and (4) adolescent women who use family planning/birth spacing may become promiscuous.

**Scored based on the responses to the following four questions: (1) felt able to start a conversation with her partner about contraception, (2) felt able to use a method of contraception even if her partner did not want her to, (3) felt able to obtain information on contraception services and products if she needed to and (4) felt able to obtain a contraceptive method if she decided to use one.

increasing age, increasing levels of education, being in education, hearing of modern contraception from interpersonal sources and in the media in last 12 months, perceiving partner and/or friend support for contraceptive use, as well as higher knowledge about contraception and self efficacy for contraception.

We found that the odds for modern contraception use were low in the respondents who perceived that their partner and/or friends did not approve of their contraception practice compared with those who perceived that their partner and/or friends did approve. Social network support has been consistently shown to influence women's decision to use contraception in various age groups and sociocultural contexts including in sub-Saharan Africa.[5 22 23] We did not observe an association between perception of mothers support for contraceptive use and use of modern contraceptives suggesting that for unmarried adolescent women, partners and friends may be more important influencers than mothers. In addition, we found that, exposure to information about contraception from interpersonal sources or in the media in last 12 months were associated with increased odds for using modern contraception. These findings call for a need for family planning programmes to target the entire community in order to raise awareness of modern contraception

and most importantly to engage male partners in support for the uptake of modern contraception.[23]

Among unmarried-sexually active women, higher knowledge and self-efficacy for contraceptive use was associated with increased odds for contraceptive use. This finding, when viewed together with other significant determinants such as advancing age and being in education, underscores the spill-over effect of girls' schooling in delaying early marriage as well as its importance in giving adolescent women more time for mental and physical maturity before embarking on sexual and reproductive roles.[9] Additionally, being in education has a potential role to overcome held misconceptions against modern contraception use.[24] This agrees with another study done in rural Mwanza which found that having low education was a risk factor for unplanned pregnancy in young women.[25]

Among unmarried women, the odds for using modern contraception were found to be significantly lower among those with one or more living children when compared with those without. This being a cross-sectional analysis, it could partly be telling that the unmarried women with living children are not using contraceptives in the first place hence risking early pregnancies. But it could also be telling us that unmarried women with living children are more likely to be young, out of school, with little exposure to information about contraception and low self efficacy to contraception, hence low contraception use.[25 26] In addition, this finding could be pointing to the negative role of mental health issues including depression facing unmarried and/or out of school teenage mothers.[26] Despite having few studies from low/middle-income countries, depression has been shown to be an independent risk factor for repeated teenage pregnancy.[27]

### Strengths and limitations

In this study, we focused on adolescent women aged 15–19 years, a population that is often excluded or underrepresented in most studies on modern contraception. We also used the probability sampling approach to interview 3511 adolescent women from 34 streets in the 15 urban and semiurban wards of Ilemela district, Mwanza. This sampling approach allows generalisation of the findings of this study to the wider population of adolescent women aged 15 - 19 years living in the surveyed wards of Ilemela district.

However, this study has some limitations that need to be noted. The cross-sectional design makes temporal causal relationships hard to establish. Also, due to small sample size of married women, we were unable to identify determinants of contraceptive use.

Lastly, we did not specifically ask the adolescent women whether they were planning on getting pregnant shortly, hence we have no data on this potential explanatory variable.

### CONCLUSION

In Northwest Tanzania, among married and unmarried-sexually active women aged 15–19 years, we found strong evidence of differences in use of modern contraceptive methods according to marital status of adolescent women.

Among unmarried-sexually active women, contraceptive use was significantly associated with increasing age, increasing levels of education, being in education, hearing information on modern contraception from interpersonal sources and in the media in the last 12 months, perceiving partner and/or friend support for contraception use, as well as higher knowledge and self efficacy for modern contraception.

In order to optimise their impact, sexual and reproductive health programmes aiming to increase uptake of modern contraceptives should consider the importance of being in education and social support for contraceptive use among adolescent women. Hence, the need to focus intervention efforts on more vulnerable unmarried-sexually active adolescent women particularly those with lower education/socioeconomic status and/or those who are already teenage mothers.

**Acknowledgements** We would like to thank all study participants for their participation and the A360 baseline study team for their dedicated work during data collection. We also thank Itad as the lead organisation responsible for the overall A360 evaluation. Avenir Health as a partner in the overall A360 evaluation. PSI Headquarters and PSI Tanzania for their support with site selection and engagement in conversation regarding the design.

**Contributors** MKN, CJA, SHK, CB and AMD were involved in conception and study design. CB provided statistical expertise. MKN and CJA were involved in drafting of the manuscript. SHK, CB and AMD were involved in critical revision of the manuscript for important intellectual content. All the authors were involved in final approval of the manuscript and decision to submit the manuscript for publication.

**Funding** The Bill & Melinda Gates Foundation and the Children's Investment Fund Foundation. The funding bodies had no role in the design of the study and collection, analysis and interpretation of data and in writing the manuscript.

**Competing interests** None declared.

**Patient consent for publication** Not required.

**Ethics approval** The study was approved by the Tanzania National Health Research Ethics review sub-committee of the National Institute for Medical Research (Ref: NIMR/HQ/R8a/Vol.IX/2549), and the research ethics committee of the London School of Hygiene & Tropical Medicine (LSHTM, Ref: 14145).

**Provenance and peer review** Not commissioned; externally peer reviewed.

**Data availability statement** Data are available upon reasonable request.

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
