## [Reviewer comments · BMJ Open]

ARTICLE DETAILS

TITLE (PROVISIONAL)	Modern contraceptive use among sexually active women aged 15 to 19 years in north-western Tanzania: results from the Adolescent 360 (A360) baseline survey
AUTHORS	Nsanya, Mussa; Atchison, Christina; Bottomley, Christian; Doyle, Aoife; Kapiga, Saidi

VERSION 1 – REVIEW

REVIEWER	Abdoulaye MAIGA Johns Hopkins University School of Public Health / USA
REVIEW RETURNED	23-Apr-2019

GENERAL COMMENTS	Main comments. This study is interesting and relevant for a few reasons. First, the study addresses a relevant topic that is the sexual and reproductive health and targets an age-specific group – adolescent girls 15-19 years – while most of the reproductive health studies are for women 15-49 years, hiding disparities and specificities for younger women. Sample size is certainly an issue, but the study also tries to highlight the difference among adolescent girls according to marital status, and sexual activity. The authors provided solid arguments and scientific background to justify the usefulness of such a study along with relevant objectives, mainly with respect to factors explaining modern contraceptive use. Unlike to what has been indicated as additional objectives, there is not enough information regarding sexuality and fertility to consider that as a main objective of the study. In contrast, mentioning the difference according to the marital status is a valid objective that can be added. Despite the sample size limitation to run a regression model for married adolescent girls, there are interesting comparison results with the unmarried group. One of the main strengths of the study is certainly the study sampling design based on probability sampling and multistage sampling. And visiting all the households in the setting is also a positive design aspect for the study. However, there is no justification in the manuscript for using a GIS sampling method. You stated that "As per study protocol, in the first eight 'streets', we randomly selected 50 GPS coordinates using ArcGIS software". Why that has been used for those eight streets? In addition, there is no clear information on the rationale for sampling 34 streets. Did you perform a sample size calculation before deciding for this number of 34 streets? If so, based on your
---

	target sample size of 3,314 adolescent girls, what was the precision or power-difference you were expecting to detect for the main indicators? Among the 15 wards surveyed, it would be desirable to know how many are urban and how many are semi-urban. You stated there were 4 rural wards in the district that were not part of the study. Why these have not been included in the study while we know that health outcomes are worse in rural setting compared to urban or semi-urban? Even if you used probability sampling, you cannot state that "This approach ensured that these findings are generalizable to the wider population of adolescent women aged 15 to 19 years living in similar urban and peri-urban wards of other regions in Tanzania where A360 intervention is being rolled out". The results are representative and valid for the study setting (Ilmela district) only, and saying that the results are generalizable to similar urban and peri-urban wards of other regions in Tanzania may be overstate. The response rate 68.6% (3,511/5,121) for eligible adolescent girls seems very low. It is great as you attempted up to three visits. What were the main reasons of absence (e.g. schooling) or was there any issue related to the definition of household member in the study? The way you presented the regression results with two models - unadjusted versus adjusted by explanatory and confounder variables – is also interesting and a good statistical analysis. P9: Concerning explanatory and confounder variables, why you did not consider whether the adolescent girl was pregnant, in post-partum amenorrhea or if she was planning to get pregnant shortly? These variables are fertility characteristic can make a difference primarily between married and unmarried adolescent girls. Moreover, why you did not use type of area (urban vs semi-urban) as a variable for adjustment, since accessibility to contraception means, being in education, access to media/information, etc. may be different depending on being in urban or semi-urban place. The proportion of married or unmarried adolescent girls may also be different according to the area of residence. The results well address the main research question that is the factors associated with contraceptive use. There are interesting results related to the impact of girls' education or social support. The huge difference in contraceptive use between unmarried (48.7%) and married (19.4%) adolescent girls is also an interesting result. However there a few comments to help improving the manuscript. Table 1. Need to clarify that first numbers are related to frequencies and seconds correspond to the percentages. P.11: Yes there is a difference overall between married and unmarried-sexually active, but it is a bit overstate saying "Overall, there was strong evidence of differences between married and unmarried-sexually active women across many of the measured
--	---

	characteristics (Table 1)". But, that may be true saying that vis-a-vis table 2. P.11. "We also observed a higher prevalence of IUCD, injectables and oral contraceptive pill use among married women". That statement is to double-check. Based on table 2, Injectables, SDM, IUCD and male condom seem instead to have higher prevalence after implant. P.21: "In a context of held misconceptions particularly against hormonal-based contraceptives for their perceived interference on fertility, this finding might reflect an attempt by unmarried sexually active women to preserve their fertility by opting for non-hormonal based and/or non-invasive methods". That could probably be a reason explaining the use of condom as main method by unmarried adolescent girls. The double role of condoms is a reason you well underlined too. There is also a need to take into account the accessibility to the methods, the exposure and frequency of sexual intercourses, and the type of sexual relationship for unmarried adolescent girls who may not need long term methods.
--	--

REVIEWER	Carolina de Vargas Nunes Coll International Center for Equity in Health, Federal University of Pelotas
REVIEW RETURNED	29-Apr-2019

GENERAL COMMENTS	This is a well written paper describing modern contraceptive use among sexually active adolescents from north-western Tanzania and its associated factors. The topic is very relevant in the context of the sustainable development agenda which aims to achieve universal access to sexual and reproductive health and, particularly, among women aged 15-19 whose progress in family planning indicators has been slower than desired. A few suggestions to improve the paper are described below. Title: Because the authors couldn't explore the factors associated with contraceptive use among all sexually active adolescents (married and unmarried) I suggest the title be changed to: "Modern contraceptive use among sexually active women aged 15 to 19 years in north-western Tanzania: results from the Adolescent 360 (A360) baseline survey". Also, revise the rest of the text to make sure this is clearly stated. Introduction. A rationale for studying contraceptive behavior separately among married and unmarried sexually active adolescents is missing. Methods. The literature remains controversial about what is considered a modern contraceptive (see Hubacher & Trussel. A definition of modern contraceptive methods. Contraception; 92 (2015) 420-421). The authors need to explain the definition of modern contraceptives adopted. The implications of including SDM and LAM as modern methods should also be discussed as these methods have questionable efficacy in the context of low and middle-income countries (where most women have low schooling). Discussion.
---

	1) The particularly low prevalence of modern contraceptive use among married adolescents should be more discussed. I suggest the authors read and incorporate the findings of a recently published manuscript exploring the contraceptive behavior of adolescents from several low- and middle-income countries according to their marital status and number of living children to their discussion. See Coll et al. Contraception in adolescence: the influence of parity and marital status on contraceptive use in 73 low and middle-income countries. Reprod Health. 2019; 16 (1). 2) The impact of losses and refusals of potentially eligible women should be discussed by the authors and included as a limitation of the study. How different were they compared to the sample studied? Is it possible that married women refused more? I suggest the authors provide the characteristics of these women for comparison.
--	---

VERSION 1 – AUTHOR RESPONSE

Reviewer: 1

Reviewer Name: Abdoulaye MAIGA

Institution and Country: Johns Hopkins University School of Public Health / USA

Please state any competing interests or state 'None declared': None declared

Please leave your comments for the authors below

Main comments.

This study is interesting and relevant for a few reasons. First, the study addresses a relevant topic that is the sexual and reproductive health and targets an age-specific group – adolescent girls 15-19 years – while most of the reproductive health studies are for women 15-49 years, hiding disparities and specificities for younger women. Sample size is certainly an issue, but the study also tries to highlight the difference among adolescent girls according to marital status, and sexual activity.

The authors provided solid arguments and scientific background to justify the usefulness of such a study along with relevant objectives, mainly with respect to factors explaining modern contraceptive use. Unlike to what has been indicated as additional objectives, there is not enough information regarding sexuality and fertility to consider that as a main objective of the study. In contrast, mentioning the difference according to the marital status is a valid objective that can be added. Despite the sample size limitation to run a regression model for married adolescent girls, there are interesting comparison results with the unmarried group.

Response: We have revised our objectives as proposed (refer to the last sentence on the last paragraph under introduction section, page 5).

One of the main strengths of the study is certainly the study sampling design based on probability sampling and multistage sampling. And visiting all the households in the setting is also a positive design aspect for the study. However, there is no justification in the manuscript for using a GIS sampling method. You stated that "As per study protocol, in the first eight 'streets', we randomly selected 50 GPS coordinates using ArcGIS software". Why that has been used for those eight streets?

Response: Due to lack of detailed list of households in the study area, we initially planned to randomly sample households using the GIS methods. This involved selecting random geographical points in each street and reaching all households within a pre-specified radius from each point to identify

people who met study eligibility. Although this approach had worked with success in a previous study we conducted in Mwanza (Kavishe B, et al. BMC Med 2015;13(1):126. doi: 10.1186/s12916-015-0357-9), in this study fewer eligible women than predicted were identified in each geographical point selected. Because of this, we decided to visit all households in the street and administered the questionnaire to all eligible and consenting women aged 15 to 19 years. We have added this clarification under the sampling strategy and sample size on page 7.

In addition, there is no clear information on the rationale for sampling 34 streets. Did you perform a sample size calculation before deciding for this number of 34 streets? If so, based on your target sample size of 3,314 adolescent girls, what was the precision or power-difference you were expecting to detect for the main indicators?

Response: Sample size and power calculations were done ahead of deciding the number of streets to sample. The estimated sample size had a 90% power to detect a 6% increment in prevalence of modern contraception use in presence of A360 intervention for 24 months. We have added this clarification under sampling strategy and sample size, page 7.

Among the 15 wards surveyed, it would be desirable to know how many are urban and how many are semi-urban. You stated there were 4 rural wards in the district that were not part of the study. Why these have not been included in the study while we know that health outcomes are worse in rural setting compared to urban or semi-urban?

Response: Of the 15 wards surveyed, 9 were urban and 6 were semi-urban. This clarification has been added under study design and setting, page 5 - 6.

We agree that health outcomes are usually worse in rural settings compared to urban and semi-urban areas, however this study (which is part of the intervention evaluation) was set in urban and semi-urban wards of Ilmela district because Population Services International (PSI) who are implementers of the A360 intervention in Tanzania are focusing their efforts in more densely populated areas.

Even if you used probability sampling, you cannot state that "This approach ensured that these findings are generalizable to the wider population of adolescent women aged 15 to 19 years living in similar urban and peri-urban wards of other regions in Tanzania where A360 intervention is being rolled out". The results are representative and valid for the study setting (Ilmela district) only, and saying that the results are generalizable to similar urban and peri-urban wards of other regions in Tanzania may be overstate.

Response: Thank you for this comment. We agree to the suggestion and we have rephrased this statement to read as follows: "Therefore, while it may not be possible to generalize our findings to the wider population of adolescent women aged 15 to 19 years living in similar urban and peri-urban wards of other regions in Tanzania, the sampling approach used allows us to generalise these findings to the wider population of adolescent women aged 15 to 19 years living in urban and semi-urban wards of Ilmela district.". Please refer to the study strengths and limitations in page 3.

The response rate 68.6% (3,511/5,121) for eligible adolescent girls seems very low. It is great as you attempted up to three visits. What were the main reasons of absence (e.g. schooling) or was there any issue related to the definition of household member in the study?

Response: The main reason of absence was schooling, either having gone for regular classes during weekdays and/or additional classes during weekends and returning home later than 6:30 p.m. or being in a boarding school. We have added this as part of the study limitations on page 3 and gave further clarification under results section in page 11.

The way you presented the regression results with two models - unadjusted versus adjusted by explanatory and confounder variables - is also interesting and a good statistical analysis.

P9: Concerning explanatory and confounder variables, why you did not consider whether the adolescent girl was pregnant, in post-partum amenorrhea or if she was planning to get pregnant shortly? These variables are fertility characteristic can make a difference primarily between married and unmarried adolescent girls.

Response: Pregnant women were not asked about contraception as they were not "at risk of pregnancy", the same applied to those in post-partum amenorrhea. So we don't have outcome data for these two subgroups. We did not specifically ask girls whether they were planning on getting pregnant shortly so we have no data on this potential explanatory variable. We have added this in page 3 and 25, under study limitations.

Moreover, why you did not use type of area (urban vs semi-urban) as a variable for adjustment, since accessibility to contraception means, being in education, access to media/information, etc. may be different depending on being in urban or semi-urban place. The proportion of married or unmarried adolescent girls may also be different according to the area of residence.

Response: We have explored the proportion of married and unmarried adolescent girls by the area of residence. There was no difference in the proportion of married and unmarried adolescent girls living in urban and semi-urban areas. We have added an additional row in Table 1 to show this, page 12. We looked further at the associations between urban/semi-urban and modern contraceptive use in unmarried sexually active girls both unadjusted, and adjusted for age and religion. There was no statistical association and therefore the variable was not included in any further models, page 17.

The results well address the main research question that is the factors associated with contraceptive use. There are interesting results related to the impact of girls' education or social support. The huge difference in contraceptive use between unmarried (48.7%) and married (19.4%) adolescent girls is also an interesting result.

However there a few comments to help improving the manuscript.

Table 1. Need to clarify that first numbers are related to frequencies and second correspond to the percentages.

Response: Clarification has been made as suggested, page 12.

P.11: Yes there is a difference overall between married and unmarried-sexually active, but it is a bit overstate saying "Overall, there was strong evidence of differences between married and unmarried-sexually active women across many of the measured characteristics (Table 1)". But, that may be true saying that vis-a-vis table 2.

Response: Thank you for this comment. We have revised the statement and used it for table 2 instead. Refer to page 12 under results section.

P.11. "We also observed a higher prevalence of IUCD, injectables and oral contraceptive pill use among married women". That statement is to double-check. Based on table 2, Injectables, SDM, IUCD and male condom seem instead to have higher prevalence after implant.

Response: Thank you for highlighting this. We have made correction to read as follows: "Of those reporting using a modern method of contraception, implants (38.5%) were the most widely used method by married women, followed by injectables (23.1%) and Standard Days Method (15.4%). Male condoms (71.6%) were the most widely used modern contraceptive method by unmarried-sexually active women, followed by Standard Days Method (15.8%)". Please refer to the third paragraph under the results section on pages 12.

P.21: "In a context of held misconceptions particularly against hormonal-based contraceptives for their perceived interference on fertility, this finding might reflect an attempt by unmarried sexually active women to preserve their fertility by opting for non-hormonal based and/or non-invasive methods". That could probably be a reason explaining the use of condom as main method by unmarried adolescent girls. The double role of condoms is a reason you well underlined too. There is also a need to take into account the accessibility to the methods, the exposure and frequency of sexual intercourses, and the type of sexual relationship for unmarried adolescent girls who may not need long term methods.

Response: Thank you for raising these important points which have been added to the manuscript on page 22 on the second paragraph under the discussion section.

Reviewer: 2

Reviewer Name: Carolina de Vargas Nunes Coll

Institution and Country: International Center for Equity in Health, Federal University of Pelotas

Please state any competing interests or state 'None declared': None declared

Please leave your comments for the authors below

This is a well written paper describing modern contraceptive use among sexually active adolescents from north-western Tanzania and its associated factors. The topic is very relevant in the context of the sustainable development agenda which aims to achieve universal access to sexual and reproductive health and, particularly, among women aged 15-19 whose progress in family planning indicators has been slower than desired.

A few suggestions to improve the paper are described below.

Title: Because the authors couldn't explore the factors associated with contraceptive use among all sexually active adolescents (married and unmarried) I suggest the title be changed to: "Modern contraceptive use among sexually active women aged 15 to 19 years in north-western Tanzania: results from the Adolescent 360 (A360) baseline survey". Also, revise the rest of the text to make sure this is clearly stated.

Response: Thank you for this suggestion. We have made changes on the title and also in text as requested by the reviewer, page 1.

Introduction. A rationale for studying contraceptive behavior separately among married and unmarried sexually active adolescents is missing.

Response: We have added a rationale in the introduction section, third paragraph, on page 5.

Methods. The literature remains controversial about what is considered a modern contraceptive (see Hubacher & Trussel. A definition of modern contraceptive methods. *Contraception*; 92 (2015) 420-421). The authors need to explain the definition of modern contraceptives adopted. The implications of including SDM and LAM as modern methods should also be discussed as these methods have questionable efficacy in the context of low and middle-income countries (where most women have low schooling).

Response: Modern contraception was defined to include the following: male and female sterilisation, contraceptive implants, intrauterine contraceptive devices (IUCD), injectables, oral contraceptive pill, emergency contraceptive pill, male condom, female condom, Standard Days Method (SDM), Lactational Amenorrhoea Method (LAM), diaphragm, spermicides, foams and jelly. We used WHO's definition (<https://www.who.int/news-room/fact-sheets/detail/family-planning-contraception>) and it is now cited on page 8 - 9 under the methods section.

Discussion.

1) The particularly low prevalence of modern contraceptive use among married adolescents should be more discussed. I suggest the authors read and incorporate the findings of a recently published manuscript exploring the contraceptive behavior of adolescents from several low- and middle-income countries according to their marital status and number of living children to their discussion. See Coll et al. Contraception in adolescence: the influence of parity and marital status on contraceptive use in 73 low and middle-income countries. *Reprod Health*. 2019; 16 (1).

Response: We have added this information in the manuscript as suggested, under discussion section, page 21.

2) The impact of losses and refusals of potentially eligible women should be discussed by the authors and included as a limitation of the study. How different were they compared to the sample studied? Is it possible that married women refused more? I suggest the authors provide the characteristics of these women for comparison.

Response: The response rate of potential eligible girls was relatively low at about 68% and the major reason for non-response was adolescent women being in school (including boarding school) and consistently arriving home later than 6:30 p.m. They were therefore more likely to be unmarried and in school. We have included this as one of our study limitations as suggested on page 3 and 24.

VERSION 2 – REVIEW

REVIEWER	Abdoulaye MAIGA Johns Hopkins University School of Public Health
REVIEW RETURNED	25-Jun-2019

GENERAL COMMENTS	The authors well addressed all the comments and suggested revisions from my first review and made changes to the manuscript accordingly. The response provided by authors with respect to the question "why you did not include the 4 rural wards in the study" is relevant and clear. The following suggestion is not really crucial, but if they can add the provided justification/response into the manuscript too, that would help clarify things for people you are going to read the paper.
---

REVIEWER	Carolina de Vargas Nunes Coll International Center for Equity in Health, Federal University of Pelotas
REVIEW RETURNED	06-Jun-2019

GENERAL COMMENTS	The authors adressed all my comments and suggestions. I recommend the manuscript be accepted.
--